# Nasal Cavity CT Imaging Contribution to the Diagnosis and Treatment of Choanal Atresia

**DOI:** 10.3390/medicina57020093

**Published:** 2021-01-21

**Authors:** Irina Šebová, Ivana Vyrvová, Jana Barkociová

**Affiliations:** 1Department of Pediatric Otorhinolaryngology, Faculty of Medicine, Comenius University and National Institute of Children’s Diseases, 83101 Bratislava, Slovakia; ivana.vyrvova@gmail.com (I.V.); jana.barkociova@gmail.com (J.B.); 2Faculty of Medicine, Masaryk University, 62500 Brno, Czech Republic

**Keywords:** choanal atresia, CT imaging, measurement of parameters

## Abstract

*Background and Objectives:* Choanal atresia is the most common congenital malformation of the nose. *Materials and Methods:* We have evaluated 24 CT images of children with choanal atresia treated at the Department of Pediatric Otorhinolaryngology FM CU and the NICD Bratislava (Slovakia). In accordance with the methodology used by Slovis et al. (1985), we have measured parameters related to anomalous development in the nasal cavity: vomer width, the width of soft atresia and the width of the air space of unilaterally developed choana. *Results:* In the group of 24 patients, 11 (46%) were male and 13 (54%) were female. The age of patients at the time of CT imaging varied. Associated syndromes had been manifested in 11 (46%) children, with 7 (29%) patients having CHARGE syndrome. In 13 (54%) cases it was a bone membranous type of atresia, in 8 (33%) cases a membranous type, and in 3 (13%) patients a bone type. Among the group of patients, unilateral disorder was present in 13 (54%) patients and bilateral in 11 (46%). Based on the Pearson’s correlation test, we have found in the studied group that the width of the vomer correlates with age, and the vomer is wider in bone atresia than in the membranous ones. Based on determining the average vomer’s width within the age groups 0–8 and >8–20, compared to the standard widths, we found that the vomer’s widths reached the upper limits of the standard ±2 SD (cm) or even exceeded that limit. The same applies to the width in soft choanal atresia. On the other hand, the width of the developed choana in the case of unilateral atresia is almost standard. *Conclusions:* The above findings are the basis for selecting the appropriate type of surgery. Currently, the gold standard is the endoscopic fenestration. associated with posterior septotomy.

## 1. Introduction

Choanal atresia is the most common congenital malformation (CM) of the nose with an incidence of 1:5000 to 1:7000 live births [1,2]. It is more common for females. Within the classification of nose CMs, according to Losee et al. (2004) [3], it falls under Type I (Table 1). In 50% of cases, it is associated with other congenital anomalies characterized by craniofacial dysmorphia, such as CHARGE syndrome, Treacher-Collins syndrome, Crouzon syndrome, Vater syndrome, Apert syndrome, Down syndrome, Edwards, Pfeiffer and Antley–Bixler syndromes [3,4,5]. Depending on the cause, the choanal atresia may be congenital or acquired; depending on its composition, it may be bone, membranous, or both; depending on the location—unilateral or bilateral, when unilaterally affected, right-sided or left-sided; and depending on the extent—complete or partial.

So far, there is no uniform theory on the cause of choanal atresia development. Hengerer and Strome (1982) [6] defined four embryological theories:(1)persistence of the buccopharyngeal membrane forming the bottom of the stomodea,(2)persistence of the Hochstetter nasobuccal membrane and failure of its perforation in the seventh week of intrauterine life,(3)abnormal occurrence of mesoderm forming adhesion in the choanal area,(4)abnormal direction of mesoderm migration (nasoseptal elements), and its excessive growth due to exogenous and local factors.

The ectopic mesenchymal tissue excess in the central part of the face forms a tangential atretic plate that extends from the vomer wall to the lateral wall of the nose. According to this theory, the excessive growth of nasoseptal elements leads to stenosis or bone atresia. Membranous atresia probably arises due to incomplete recanalization of the epithelial plug in the 13th to 20th week of gestation. Other theories of atresia include the Pasquini (2003) [7] and Lalani theory (2004) [8]:(5)persistence of epithelial cells in the nasal cavity that proliferate between the sixth and eighth week of gestation,(6)excessive growth of lamina horizontalis and lamina perpendicularis ossis palatini.

The theory of persistence of the nasobuccal membrane (nasal sacs’ bottom) is the most often mentioned in the literature. Nevertheless, it seems unlikely, as the nasobuccal membrane forms near the outer nostrils, and after its perforation, primitive choanae start to form. However, the atresia of choanae probably occurs only in the later stages of development of a definitive secondary choanae. Due to the presence of associated anatomical abnormalities of the nasal cavity and choanal area, which usually result in a narrowing throughout half of the nasal cavity on the atresia side, the most probable cause is a generalized disorder of mesenchymal cell migration that also affects the development of the base of the skull [9].

Present diagnostic methods (endoscopy and CT imaging) have remarkably pushed our knowledge about anatomical features in each patient further. It makes the diagnosis of choanal atresia clear (type, thickness of the atretic segment). Typical changes in the nasal cavity on the atretic side include the thickening of the posterior part of the vomer, the thickening of the medial plate of the processus pterygoideus and the medialization of the lateral nasal wall on the level of the choana [2]. The aim of our study was to find out if the CT imaging evaluation with respect to the parameters followed in patients with choanal atresia will help us determine the optimal surgical treatment.

## 2. Materials and Methods

We have evaluated the nasal cavity CT imaging of 24 pediatric patients diagnosed with ICD-10: Q30.0 choanal atresia examined at the Department of Pediatric Otorhinolaryngology of the Medical Faculty of the Comenius University and the National Institute of Children’s Diseases in Bratislava (Slovakia). We have recorded the sex of patients, the presence of syndromes, their age at the time of the nasal cavity CT examination, and the types of atresia (congenital–acquired, unilateral–bilateral). In the case of the unilateral atresia, we have examined which half of the nasal cavity was affected. We determined the character of the atretic plate (membranous, bone-membranous, and bone types) and their locations (unilateral, bilateral) [4]. Using the method of Slovis et al. (1985) [10], a CT section paralleling the posterior hard palate at the level of the pterygoid plates was obtained in all children (Figure 1). The sections were 0.5 to 1 mm thick. The TomoCon PACS system was used for CT scanning. The measurements were obtained by one examiner with Tomocon Viewer application measuring tools.

We measured the width of the vomer (V) at the posterior end, and the width of the air space of the choanae or membranous atresia based on the measurement of the dimension from the lateral wall of the nasal cavity to the edge of the vomer (LWNC-V) of its posterior end (Figure 2). The obtained results were compared with the standards of these parameters according to the Slovis et al. (1985) [10] (Table 2) and statistically processed using the IBM SPSS Statistics for Macintosh, Version 27.0. Armonk, NY: IBM Corp.

In patients with bone or bone-membranous atresia, the width of LWNC-V is non-measurable.

## 3. Results

In a group of 24 (100%) patients with a basic diagnosis made according to ICD-10 Q30.0 Choanal atresia (Table 3), there were 11 (46%) male and 13 (54%) female patients. The age at the time of CT imaging was as follows: neonatal in 11 (45.7%) cases; 7 (30%) children were from 1 month to 2 years old; 1 child between 2 and 4 years of age (4%); 1 child aged from 2 to 4 years (4%); 1 child between 14–16 years (4%); another 2 (8.3%) patients were 16–18 years old; 1 patient in 18–20 years age group (4%). The associated syndromes were present in 11 children (46%), with 7 patients (29%) having CHARGE syndrome, the remaining four patients having Vacterl syndrome, Rubinstein–Taybi syndrome, Treacher-Collins syndrome, and Down syndrome. Two families, where we assumed the presence of CHARGE syndrome, refused to take genetic tests. All identified cases of atresia were congenital and fully developed. During the CT examinations of the nasal cavity, we found that 13 (54%) of cases represented a bone-membranous type of atresia, in 8 cases (33%) it was a membranous type of atresia and in 3 patients (13%) it was a bone type of atresia. Thirteen (54%) patients were unilaterally affected, and 11 (46%) patients bilaterally. Of the 13 (100%) patients with unilateral choanal atresia, 8 were left-sided (62%) and 5 right-sided (38%).

To evaluate the vomer width measurement results, we compared the individually measured vomer width with the mean ± 2 SD in accordance with Slovis et al. (1985) [10] (Figure 3). We found, in the age group of 0–8 years with 20 patients (patients 1–20), that the average vomer width ± 2 SD in the whole group was 0.407 (0.305–0.51) cm, significantly exceeding the norm. In the group of 4 patients aged >8–20 (patients 21–24), two had unilateral choanal atresia and two bilateral choanal atresia. It can be seen from the diagram that all patients have a vomer width ± 2 SD at the upper limit. In this group, we also found that the average vomer width ± 2 SD is greater than the mean of 0.535 (0.474–0.59) cm.

We continued measuring the LWNC-V width (lateral wall nasal cavity–vomer) (1) on the side of the membranous atresia at the end of the nasal cavity (Figure 4), and (2) on the side of a unilaterally developed choana at the same level (Figure 5). The LWNC-V width in patients with membranous choanal atresia (*n* = 8) aged 0 to 2 years of age was 0.363 ± 0.291–0.435, which is significantly less than the standard. The LWNC-V width in patients with unilaterally developed choana was 0.639 ± 0.482–0.796, which is close to the norm and does not exceed its upper limit.

Based on the Pearson correlation test between the vomer width and the age within the observed group of patients with the choanal atresia (*n* = 24), we found a positive correlation between the vomer width and the age. There is the same linear correlation as in the normal population, i.e., in children with a standard finding in the nasal cavity. The older the patient, the wider the vomer, even in patients with choanal atresia (r = 0.442, *p* = 0.031, 95% CI 0.108–0.760, SE 0.165). Besides, based on the Pearson correlation test, between the vomer width and the presence of the unilateral or bilateral choanal atresia in the group of 0–2 aged children (n = 20), we found that in the bilateral choanal atresia, the vomer is on average wider than in the unilateral atresia (r = −0.578, *p* = 0.012, 95% CI 0.274–0.807, SE 0.136).

## 4. Discussion

Anderhuber (2010) [11] states that the rupture of the membrana oronasalis occurs in the fifth/sixth embryonic week when the primary choana is formed. Subsequently, a secondary palatum is created from the tissue behind the foramen incisivum, definitively separating the nasal and oral cavities, and creating the final shape of the choana. For this reason, the most common theory for the development of choanal atresia is the notion of membrana oronasalis persistence. The evaluated CT images of patients with choanal atresia confirm that it is the case of a luminescence disorder of various degrees in the area of the presumed choana with regard to the time of the disorder’s onset—from prominent bone plate to bone-membrane plate to membranous plug, between the fifth and twentieth week of embryo development, according to the available literature [12,13].

The causes of choanal atresia are still not clear. Retinoic acid deficiency is discussed as it affects the ontogenesis and homeostasis of many tissues. It causes the overexpression of FGF-8, which could lead to the persistence of the nasal valve. Other authors consider the possible effect of thioamides used during pregnancy. Kurosaka (2018) focuses on the role of genes involved in 50% in the development of choanal atresia. They are usually part of the signaling pathways that control cellular activity and are essentially involved in the normal or abnormal development of the neural plate, for example via the fibroblast growth factor (FGF). Syndromes characterized by aberrant development of the cranial part of the neural plate are referred to as neurocristopathies; these include CHARGE syndrome, Treacher-Collins and DiGeorge syndrome, and others. Another group includes syndromes caused by FGF mutations, such as Crouzon and Pfeiffer syndromes. Furthermore, there are syndromes characterized by accelerated bone growth, for example in craniodiaphyseal dysplasia or Marshall syndrome, then syndromes typical for middle facial disorders, such as Fraser syndrome, and finally syndromes caused by mutations in the SHH (sonic hedgehod) signaling pathway, such as Pallister–Hall syndrome [14].

Hengerer et al. (2008) [6] underline the importance of the correct identification of obstruction locations in the nasal cavity, as they may represent limits to the success of surgical treatment. They draw attention to the “2–1” rule, which relates to the predominance of unilateral over bilateral choanae atresia, the predominant female sex, and the more frequent right-sided occurrence of unilateral atresia. We did not identify such a typical representation in the study group—unilateral atresia was 54%, the representation of the female sex was also 54%, and on the contrary, left-sided atresia predominated (62%) in the cases of unilateral atresia, but our results are limited by the small count in the group. According to the literature, choanal atresia is at approximately 50% associated with the presence of syndromes, which corresponds to our result of 46%. The most frequent is CHARGE syndrome; Lofty and Al-Noury (2011) [15] state that 75% of 9 patients had CHARGE syndrome in their group. Burrow et al. (2009) [16] identified a syndrome in 51% of 129 patients, predominantly CHARGE syndrome. These findings correlate with our findings. In our group of 11 patients, 7 children (29%) had the syndrome, and the two families, where presence had been expected, refused to undergo a genetic examination. In his review of choanal atresia, Kwong (2015) [17] refuted the historical fact that 90% of cases are bone atresia, and only 10% are membranous type. Based on a retrospective evaluation of CT images and histological examination of the findings in 63 patients, he determined that the bone atresia was present in only 29% of patients and the others had a mixed syndrome form of atresia, which was present in 71% cases. In our group of patients, the majority had bone-membranous atresia as well (*n* = 13, 54%).

Ginat and Robson (2015) [18] state that in children under 2 years of age, the width of the postero-inferior part of the vomer is ≤2.3 mm and the diameter of the choana is ≥3.7 mm, while its width increases by 0.09 mm/year. They compared the nasal cavity CT examinations of 11 patients with choanal atresia with a control group of 66 patients with normal findings in the nasal cavity. In our work, we followed the standards set out by Slovis et al. (1985) [10]. For neonates, they found that the space between the lateral wall of the nasal cavity and the vomer was 0.67 mm, increasing by 0.27 mm each year until the age of 20. The average vomer thickness was 2.3 mm in patients under 8 years of age and 2.8 mm in patients aged from 8 to 20 years. The thickness of bone atresia varied from 1 to 12 mm, depending on bone changes in the area of the lamina pterygoidea medialis. In our group, we found in all examined parameters that the width of the vomer, LWNC-V, and the width of the developed choana in unilateral choanal atresia are on average below the standards set for a given age by Slovis et al. (1985) [10]. Aslan (2009) [19] examined 17 different parameters on HRCT nasal images in 9 children with bilateral choanal atresia, compared to the control group of 104 pediatric patients. They found that in the presence of atresia, the nasal cavity is narrowed to the full extent. They did not find statistically significant differences in the length of the nasal septum, or in the area of the nasopharynx.

Lofty and Al-Noury (2011) [15] based on CT examinations of the nasal cavity compared to the group of 9 pediatric patients, showed its great informative value in the diagnosis of choanal atresia, as well as in the detection of associated maxillo-facial anomalies, including stenosis of the apertura piriformis, which is a typical manifestation of CHARGE syndrome. When evaluating CT images of the nasal cavity, we confirmed the presence of typical anatomical changes accompanying choanal atresia, such as the thickening of the posterior part of the vomer, medialization of the lateral nose wall at the level of the medial plate processus pterygoideus and narrower dimensions of the choana with the membranous atresia, as well as on the unaffected side in unilateral atresia. The same changes are reported by several authors [6,9].

Members of IPOG (International Pediatric Otolaryngology Group) from eight countries (Australia, Canada, France, Ireland, Italy, Portugal, United Kingdom and United States) have agreed on the importance of HRCT nasal cavity examination as the gold standard in the diagnosis and treatment of choanal atresia and evaluated the experience of 22 tertiary centers in the consensual document of 2019. Regarding treatment, 89.3% of participants prefer an endoscopic approach to the treatment of choanal atresia; the transpalatal approach is used in exceptional cases [20]. It also follows from our results that in the case of separate transnasal curettage for the purpose of fenestration of atretic choanae, the scope of intervention is insufficient, due to the above-mentioned changes. It does not allow the removal of excess bone in the area of the choanal atresia to a sufficient extent and leaves an enlarged back part of the vomer in place. Currently, we can achieve an adequate extent of the fenestration by utilizing a microendoscopic technique with the use of a drill or a bone shaver. This will remove the excess bone in the area of the atretic plate and any membranous part of the choana atresia. A laser should be used to prevent bleeding. Subsequently, the obtained space is enlarged using a posterior septostomy, removing the dorsal excessively thickened part of the vomer. Rajan and Tunkel (2018) [2] underline the importance of covering the wound areas in the neochoana with mucosal grafts using fibrin glue.

The safety of surgery can be increased by the IGNS (Image Guided Navigation Surgery). In the case study by Ji et al. (2020) [5], they report successful use of the IGNS in the surgical endoscopic treatment of an unrecognized complex bilateral choanal atresia, associated with significant deviation of the nasal septum, and with the Tessier 3 facial cleft.

## 5. Conclusions

Contrary to the recommendation by Slovis et al. (1985) [10] to indicate a transpalatal approach for the bilateral bone atresia, and endoscopically address only unilateral choanal atresia, based on the current modern endoscopic options, we have evidently moved to a transnasal microendoscopic approach [12,20,21]. The consistent fenestration of the atretic choana is combined with the posterior nasal septotomy. The results of our study strongly contributed to the justification of this procedure; it allows the creation of a spacious neochoana for both halves of the nasal cavity without the need for long-term stenting. The transpalatal approach is indicated only for very special and complicated cases.

## Figures and Tables

**Figure 1 medicina-57-00093-f001:**
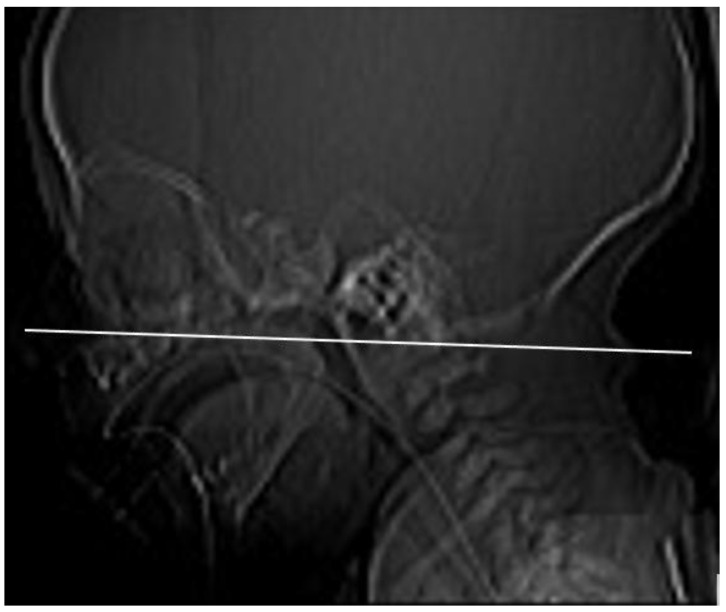
The measurement plane: CT section paralleling the posterior hard palate at the level of the pterygoid plates.

**Figure 2 medicina-57-00093-f002:**
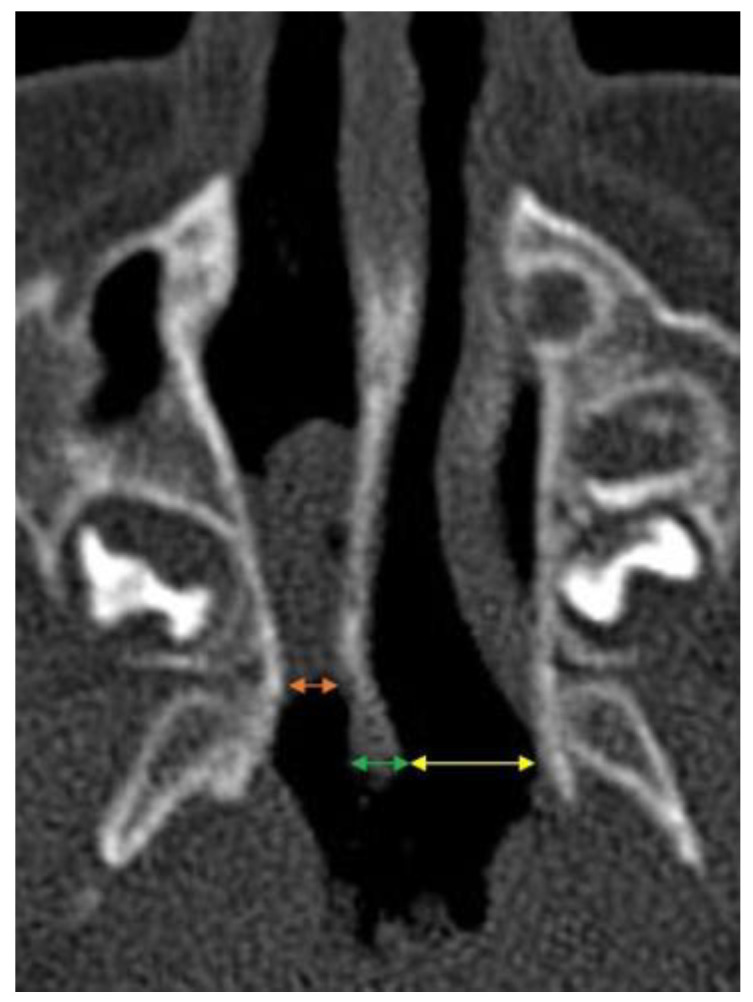
The measurement method: the lateral wall of the nasal cavity to the edge of the vomer (LWNC-V) of the membranous atresia (orange arrow), the width of the vomer (green arrow) and the LWNC-V of the choanal air space (yellow arrow).

**Figure 3 medicina-57-00093-f003:**
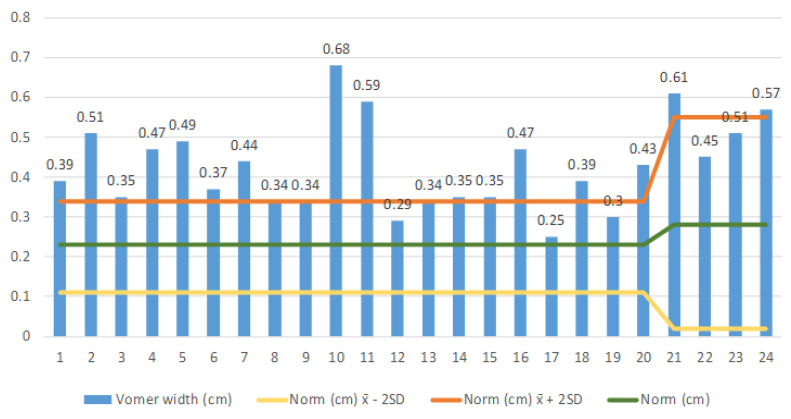
The results of the comparison of individual vomer width in the group of patients with choanal atresia (*n* = 24) with the norm ± 2 SD according to Slovis et al. (1985) [10]. Key: yellow colour–norm-2 SD, green colour–norm, orange colour–norm + 2 SD in patients, blue colour–an individual value of each patient.

**Figure 4 medicina-57-00093-f004:**
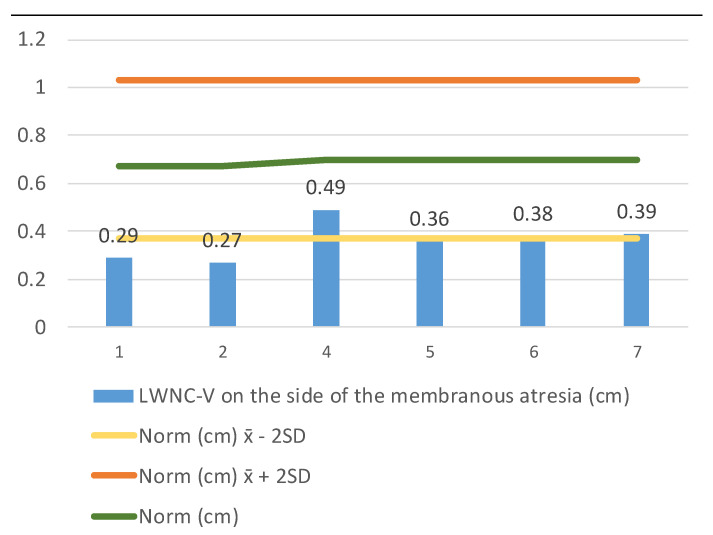
Comparison of the norm ± 2 SD for the choana size according to Slovis et al. (1985) [10] and LWNC-V width in patients with membranous choanal atresia (*n* = 7) aged 0–2 years. Key: yellow color–norm-2 SD, green color–norm, orange color–norm + 2 SD in patients, blue color–an individual value of each patient.

**Figure 5 medicina-57-00093-f005:**
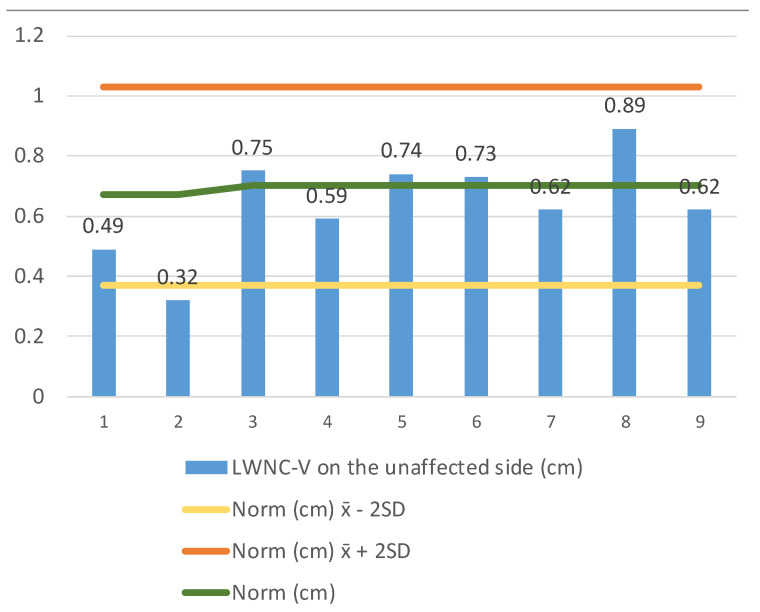
Comparison of the norm ± 2 SD for the choana size according to Slovis et al. (1985) [10]. LWNC-V of the developed choana in unilateral membranous choanal atresia in the observed group (*n* = 9) of 0–2 years age. Key: yellow color–norm-2 SD, green color–norm, orange color–norm + 2 SD in patients, blue color–an individual value of each patient.

**Table 1 medicina-57-00093-t001:** Classification of congenital malformations of the nose according to Losee et al. (2004) [3].

Type I–Hypoplasia and atrophy (62%)	Half nose nasal aplasia, missing nose (arhinia), absence of some nasal structures (often part of craniofacial malformations), stenosis, apertura piriformis, choanal atresia, and stenosis
Type II–Hyperplasia and duplicity (1%)	Proboscis lateralis, doubling of the nose and doubling of the nostril
Type III–Clefts (16%)	Clefts of the nose
Type IV–Neoplasia (20%)	Encephaloceles, gliomas, dermoid cysts, fistulas, and vascular anomalies

**Table 2 medicina-57-00093-t002:** The standard dimensions of the choanae and vomer by patients’ age categories according to Slovis et al. (1985) [10].

Measurement	Age (Years)	Mean (cm)	±2 Standard Deviations (cm)
**Choanal air space**	Birth	0.67	0.34–1.01
	>0–2	0.70	0.37–1.03
	>2–4	0.75	0.42–1.09
	>4–6	0.80	0.47–1.14
	>6–8	0.86	0.53–1.19
	>8–10	0.91	0.58–1.25
	>10–12	0.97	0.63–1.30
	>12–14	1.02	0.69–1.35
	>14–16	1.07	0.74–1.41
	>16–18	1.13	0.79–1.46
	>18–20	1.18	0.85–1.51
**Vomer width**	0–8	0.23	0.11–0.34
	>8–20	0.28	0.02–0.55

**Table 3 medicina-57-00093-t003:** The characteristics of the group (*n* = 24) taking into account the sex, age at the time of the nasal cavity CT examination, presence of the syndrome, type of atresia according to its composition, and, in the case of unilateral atresia, the lateral location.

Number	Sex	Age	Type of Choanal Atresia	Vomer Width	LWNC-V on the Side of Soft Atresia (cm)	LWNC-V on the Unaffected Side (cm)	Syndrome
1	F	Nb/2 d	M–bilat.	0.39	0.29/0.29	-	CHARGE
2	F	Nb/2 d	B–bilat.	0.51	-	-	
3	M	Nb/3 d	BM–unilat./left	0.35	-	0.49	
4	M	Nb/4 d	BM–bilat.	0.47	-	-	Treacher-Collins
5	M	Nb/4 d	BM–bilat.	0.49	-	-	Vacterl
6	M	Nb/6 d	BM–unilat./right	0.37	-	0.32	Rubinstein–Taybi
7	F	Nb/7 d	BM–bilat.	0.44	-	-	
8	F	Nb/7 d	BM–bilat.	0.34	-	-	CHARGE
9	M	Nb/10 d	M–bilat.	0.34	0.27/0.31	-	CHARGE
10	F	Nb/11 d	B–bilat.	0.68	-	-	Down
11	M	Nb/19 d	BM–bilat.	0.59	-	-	
12	M	1.5 m	M–unilat./left	0.29	0.49	0.75	
13	F	2 m	M–unilat./left	0.34	0.36	0.59	
14	M	2.5 m	M–unilat./left	0.35	0.38	0.74	CHARGE
15	F	12 m	M–unilat./right	0.35	0.39	0.73	CHARGE
16	F	14 m	BM–unilat./left	0.47	-	0.62	
17	F	20 m	BM–unilat./left	0.25	-	0.89	
18	M	24 m	B–unilat./left	0.39	-	0.62	
19	F	3 y	BM–unilat./right	0.30	-	0.74	
20	F	4.5 y	BM–unilat./right	0.43	-	0.81	
21	M	15 y	BM–bilat.	0.61	-	-	CHARGE
22	F	16 y	M–unilat./right	0.45	0.54	0.9	
23	M	16 y	BM–bilat.	0.51	-	-	CHARGE
24	F	19 y	BM–unilat./left	0.57	-	0.71	

F = female, M = male, Nb = newborn, d = days, m = months, y = years, B = bony, M = membranous, BM = bony-membranous, bilat. = bilateral, unilat. = unilateral, LWNC-V = lateral wall of nasal cavity-to-vomer measurement.

## Data Availability

All the data are available in this article (Table 3).

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
