# Peer review of "Nasal Cavity CT Imaging Contribution to the Diagnosis and Treatment of Choanal Atresia"

_medicina, 2021, doi:10.3390/medicina57020093_

Round 1
Reviewer 1 Report
Thank you for getting the opportunity to review this interesting work. The study deals with the important pediatric topic of choanal atresia and provides measurements in this regard that are scarce in literature. Therefore, the study has a clear academic merit and deserves to be published in Medicina. I recommend to improve a couple of points as shown below, which predominantly should help to better present the work for the interested readership. I wish the authors all the best with future research on the topic. Kind regards.
Language
I recommend that the authors ask a native speaker to proofread the manuscript or use one of the freely available online products (e.g., Grammarly.com) to check the grammar of the manuscript. There are a couple of inaccuracies throughout the manuscript, which should be omitted. A couple of examples are given elow:
Line 20: “patients” instead of “patient”
Lines 22/24: “atresia” instead of “Atresia”
Line 33: “females” instead of “girls”
Line 80: Please use “sex” instead of “gender” and do so throughout the manuscript
Introduction
Line 45: The authors present the thickening of the vomer as a typical change of the nasal cavity at the atresia side. Please provide an appropriate reference for this fact as well as the other ones mentioned in this sentence.
Lines 58-60: Please reference the statements made.
General comments on the introduction: The authors nicely introduce atresia as the most common congenital malformation of the nose and present current theories of its development. However, it remains unclear so far to the reader what the authors intended to find out additionally with the given work. Which hypothesis was stated that/what was the objective to undertake the current observations? Please implement this at the end of the introduction, which will largely increase the understanding of the valuable data the authors have obtained.
Materials and Methods
The authors defined the measurements that were taken and reference Slovis et al. (1985). One thing that comes to mind when looking at the Figures 1 and 2 is that they were acquired at different heights in a sagittal plane. Could the authors define the exact comparable plane that was chosen for all individuals? If sagittal heights were selected randomly, this has to be added at least as an important limitation at the end of the discussion. In the discussion, please refer to how other cited studies that used CT images for the here given purpose dealt with the sagittal plane (the height the images were taken).
Results
Lines 142: The authors stated that ages were divided from 0-8 years and 8-20 years but that means 8 years would be in both categories (even though not of practical relevance according to the ages of the individuals). I recommend to change it to 0-8 and >8-20 years (as done by Slovis 1985).
Figure 3 and 4: I recommend to also include a line of the norm not only its doubled SDs
Lines 159 to 161: “We have continued with measuring the LWNC-V width (lateral wall nasal cavity – 159 vomer) 1) on the side of the membranous atresia at the end of the nasal cavity (Figure 5), 160 and 2) on the side of a unilaterally developed choana at the same level (Figure 6).” This sentence should be part of the materials and methods or changed to clearly express that the results of these measurements are shown in the respective figures (or both). Also, LWNC-V should be properly introduced not only in the text but also in the figures where it is used. The reader cannot be expected to know this abbreviation.
Discussion
Line 231: It should be “2.3” instead of “2,3”
Lines 239-242: “In our group, 239 we found in all examined parameters that the width of the vomer, LWNC-V and the width 240 of the developed choana in unilateral choanal atresia is on average below the standards 241 set for a given age by Slovis et al. (1985) [10].” How do the authors interpret these findings? What does it mean in their opinion and how can the observed data be used clinically (as claims were made in the abstract that the here presented values would be the basis to select the appropriate type of surgery)?
Lines 263-267: “It also follows from our results that in the case of separate transnasal curettage for the purpose of fenestration of atretic choanae the scope of intervention is insufficient, due to the above-mentioned changes. It does not allow the removal of excess bone in the area of the choanal atresia to a sufficient extent and leaves an enlarged back part of the vomer in place.” It is unclear how the results of this study lead to this conclusion, please specify.
Conclusions
The conclusions are not supported by the presented results. Please provide a conclusion, which can be drawn from the stated results.
Author Response
Open Review
(x) I would not like to sign my review report
( ) I would like to sign my review report
English language and style
( ) Extensive editing of English language and style required
(x) Moderate English changes required
( ) English language and style are fine/minor spell check required
( ) I don't feel qualified to judge about the English language and style
Yes |
Can be improved |
Must be improved |
Not applicable |
|
Does the introduction provide sufficient background and include all relevant references? |
( ) |
(x) |
( ) |
( ) |
Is the research design appropriate? |
(x) |
( ) |
( ) |
( ) |
Are the methods adequately described? |
( ) |
(x) |
( ) |
( ) |
Are the results clearly presented? |
(x) |
( ) |
( ) |
( ) |
Are the conclusions supported by the results? |
( ) |
( ) |
(x) |
( ) |
Comments and Suggestions for Authors
Thank you for getting the opportunity to review this interesting work. The study deals with the important pediatric topic of choanal atresia and provides measurements in this regard that are scarce in literature. Therefore, the study has a clear academic merit and deserves to be published in Medicina. I recommend to improve a couple of points as shown below, which predominantly should help to better present the work for the interested readership. I wish the authors all the best with future research on the topic. Kind regards.
Point 1: Moderate English changes required
I recommend that the authors ask a native speaker to proofread the manuscript or use one of the freely available online products (e.g., Grammarly.com) to check the grammar of the manuscript. There are a couple of inaccuracies throughout the manuscript, which should be omitted. A couple of examples are given elow:
Line 20: “patients” instead of “patient”
Lines 22/24: “atresia” instead of “Atresia”
Line 33: “females” instead of “girls”
Line 80: Please use “sex” instead of “gender” and do so throughout the manuscript
Response 1: We have done English language correction with help of the company LEXMAN and the online corrector grammarly.com. Minor errors (Atresia – atresia,...) are eliminated.
Point 2:
Does the introduction provide sufficient background and include all relevant references? |
( ) |
(x) |
Response 2: We have explained the aim of the study more precisely.
Point 3: Introduction Line 45: The authors present the thickening of the vomer as a typical change of the nasal cavity at the atresia side. Please provide an appropriate reference for this fact as well as the other ones mentioned in this sentence.
Response 3: We have supported the typical changes in the nasal cavity with the appropriate reference.
Point 4: Lines 58-60: Please reference the statements made.
Response 4: These hypotheses are from Hengerer and Strome (1982), reference 6.
Point 5: General comments on the introduction: The authors nicely introduce atresia as the most common congenital malformation of the nose and present current theories of its development. However, it remains unclear so far to the reader what the authors intended to find out additionally with the given work. Which hypothesis was stated that/what was the objective to undertake the current observations? Please implement this at the end of the introduction, which will largely increase the understanding of the valuable data the authors have obtained.
Response 5: We have changed Introduction with the accent on the surgical exploitation of data.
Point 6: Materials and Methods
The authors defined the measurements that were taken and reference Slovis et al. (1985). One thing that comes to mind when looking at the Figures 1 and 2 is that they were acquired at different heights in a sagittal plane. Could the authors define the exact comparable plane that was chosen for all individuals? If sagittal heights were selected randomly, this has to be added at least as an important limitation at the end of the discussion. In the discussion, please refer to how other cited studies that used CT images for the here given purpose dealt with the sagittal plane (the height the images were taken).
Response 6: We have modified Methods with the accent on the method from Slovis et al. (1985) with new pictures from our study (measurement plane and method).
Point 7: Results
Lines 142: The authors stated that ages were divided from 0-8 years and 8-20 years but that means 8 years would be in both categories (even though not of practical relevance according to the ages of the individuals). I recommend to change it to 0-8 and >8-20 years (as done by Slovis 1985).
Response 7: We have accepted this correction
Point 8: Figure 3 and 4: I recommend to also include a line of the norm not only its doubled SDs
Response 8: We have created one figure from Figures 3 and 4 and we have included a line of the norm into the figure.
Point 9: Lines 159 to 161: “We have continued with measuring the LWNC-V width (lateral wall nasal cavity – 159 vomer) 1) on the side of the membranous atresia at the end of the nasal cavity (Figure 5), 160 and 2) on the side of a unilaterally developed choana at the same level (Figure 6).” This sentence should be part of the materials and methods or changed to clearly express that the results of these measurements are shown in the respective figures (or both). Also, LWNC-V should be properly introduced not only in the text but also in the figures where it is used. The reader cannot be expected to know this abbreviation.
Response 9: We changed basically Methods with the new description of the measurement plane and the measurement method. Also we applied new pictures.
Point 10: Discussion Line 231: It should be “2.3” instead of “2,3”
Response 10: Correction done.
Point 11: Lines 239-242: “In our group, 239 we found in all examined parameters that the width of the vomer, LWNC-V and the width 240 of the developed choana in unilateral choanal atresia is on average below the standards 241 set for a given age by Slovis et al. (1985) [10].” How do the authors interpret these findings? What does it mean in their opinion and how can the observed data be used clinically (as claims were made in the abstract that the here presented values would be the basis to select the appropriate type of surgery)?
Response 11: From our results we know that each space, which we are able to win with the fenestration and posterior nasal septotomy is important for breathing. The diameter of the developed choana is at unilateral choanal atresia normal, but together with the minor diameter of the fenestra diameter after surgery on the other side a neochoana is smaller than normal. It is very important to combine consistent fenestration with the posterior nasal septotomy.
Point 12: Lines 263-267: “It also follows from our results that in the case of separate transnasal curettage for the purpose of fenestration of atretic choanae the scope of intervention is insufficient, due to the above-mentioned changes. It does not allow the removal of excess bone in the area of the choanal atresia to a sufficient extent and leaves an enlarged back part of the vomer in place.” It is unclear how the results of this study lead to this conclusion, please specify.
Response 12: see response 11
Point 13: Conclusions The conclusions are not supported by the presented results. Please provide a conclusion, which can be drawn from the stated results.
Response 13: We changed Conclusions with the accent on the surgical procedure needed for successful reconstruction of the nasal cavity in patients with choanal atresia.
Submission Date
31 December 2020
Date of this review
05 Jan 2021 23:47:09
© 1996-2021 MDPI (Basel, Switzerland) unless otherwise stated
Reviewer 2 Report
This is a fine piece of clinically applied research with a thorough anatomical focus. A few minor changes are required, however, I can see that this manuscript draft well combines embryological research questions with clinically-relevant symptoms. Please assure to proof-read the paper for English typos etc.
Minor errors: Change Atresia throughout to atresia, the same applies to Choana, Vomer
Add more information on the devices and programs deployed (e.g., place of manufacture, lines 90-92
Author Response
Open Review
( ) I would not like to sign my review report
(x) I would like to sign my review report
English language and style
( ) Extensive editing of English language and style required
( ) Moderate English changes required
(x) English language and style are fine/minor spell check required
( ) I don't feel qualified to judge about the English language and style
Yes |
Can be improved |
Must be improved |
Not applicable |
|
Does the introduction provide sufficient background and include all relevant references? |
(x) |
( ) |
( ) |
( ) |
Is the research design appropriate? |
(x) |
( ) |
( ) |
( ) |
Are the methods adequately described? |
(x) |
( ) |
( ) |
( ) |
Are the results clearly presented? |
(x) |
( ) |
( ) |
( ) |
Are the conclusions supported by the results? |
(x) |
( ) |
( ) |
( ) |
Comments and Suggestions for Authors
This is a fine piece of clinically applied research with a thorough anatomical focus. A few minor changes are required, however, I can see that this manuscript draft well combines embryological research questions with clinically-relevant symptoms. Please assure to proof-read the paper for English typos etc.
Minor errors: Change Atresia throughout to atresia, the same applies to Choana, Vomer
Add more information on the devices and programs deployed (e.g., place of manufacture, lines 90-92
Submission Date
31 December 2020
Date of this review
08 Jan 2021 11:26:31
© 1996-2021 MDPI (Basel, Switzerland) unless otherwise stated
Point 1: English language and style are fine/minor spell check required
Response 1: We have done English language correction with help of the company LEXMAN and the online corrector grammarly.com. Minor errors (Atresia – atresia,...) are eliminated.
Point 2: Add more information on the devices and programs deployed (e.g., place of manufacture, lines 90-92
Response 2: We changed basically Methods with the new description of the measurement plane and the measurement method. Also we have used new pictures.
Reviewer 3 Report
Šebová et al. reported their experience of using nasal cavity CT in the diagnosis of childhood. choanal atresia. At the end of the study, they enrolled 24 cases and identified almost equal numbers of patients with unilateral or bilateral abnormalities. The results also showed the width of the Vomer correlating with age, and the wider Vomer was found in bone Atresia than in the membranous ones. They concluded an important value of CT in assisting surgical plan for choanal atresia. Overall, the manuscript is well written, albeit with several grammatical errors making some descriptions ambiguous. Besides, the methodology presentation should be improved to convey more clear information and future recapitulation. I have several comments as follows.
- Introduction Line 36, the authors should spell out the “CHARGE syndrome.”
- The source data referred by Table 1 is too old (published in 2004). The authors should try to cite more updated data regarding the classification.
- I do not think Table 2 is necessary to be shown in the manuscript since it is just reference data for the patients, not extracted data from the authors’ patients. Besides, the criteria are also a bit old. Is there any revision of the criteria?
- The presentations of Figures 3-6 are disorganized and somehow confusing to readers. The presentation of the SD bar is different from the standard display of scientific articles. I strongly suggest that the authors reorganize Figures 3-6 and consider simplifying them into 2 figures.
- Line 194-207, since this paper did not present any data regarding the gene anomalies regarding choanal atresia, I do not think the discussion of genopathy is necessary for this paragraph.
- Line 267-273, as the main purpose of this paper, stated in the title is the value of applying nasal cavity CT; however, no data is presenting the surgical procedure related to CT findings. Therefore, I think it would be inappropriate to state surgical intervention in the discussion.
Author Response
Open Review
(x) I would not like to sign my review report
( ) I would like to sign my review report
English language and style
( ) Extensive editing of English language and style required
(x) Moderate English changes required
( ) English language and style are fine/minor spell check required
( ) I don't feel qualified to judge about the English language and style
Yes |
Can be improved |
Must be improved |
Not applicable |
|
Does the introduction provide sufficient background and include all relevant references? |
( ) |
(x) |
( ) |
( ) |
Is the research design appropriate? |
(x) |
( ) |
( ) |
( ) |
Are the methods adequately described? |
( ) |
(x) |
( ) |
( ) |
Are the results clearly presented? |
( ) |
( ) |
(x) |
( ) |
Are the conclusions supported by the results? |
( ) |
(x) |
( ) |
( ) |
Comments and Suggestions for Authors
Šebová et al. reported their experience of using nasal cavity CT in the diagnosis of childhood. choanal atresia. At the end of the study, they enrolled 24 cases and identified almost equal numbers of patients with unilateral or bilateral abnormalities. The results also showed the width of the Vomer correlating with age, and the wider Vomer was found in bone Atresia than in the membranous ones. They concluded an important value of CT in assisting surgical plan for choanal atresia. Overall, the manuscript is well written, albeit with several grammatical errors making some descriptions ambiguous. Besides, the methodology presentation should be improved to convey more clear information and future recapitulation. I have several comments as follows.
Point 1: Language – moderate English changes required.
Response 1: We have made English language correction with the help of the company LEXMAN and the online corrector grammarly.com.
Point 2: Introduction Line 36, the authors should spell out the “CHARGE syndrome.”
Response 2: The abbreviation CHARGE is used officially in books, publications, at the website of CHARGE SYNDROM FOUNDATION, MedlinePlus Genetics, Rare Disease Database. Please, is it possible to accept it or do we need to change it?
Point 3: The source data referred by Table 1 is too old (published in 2004). The authors should try to cite more updated data regarding the classification.
Response 3: We have found in the literature that this concrete classification is mostly accepted, e.g. see Fijalkowska, M., Antoszewski, B.: Classification of congenital nasal deformities: a proposal to amend the existing classification. European Archives of Oto-Rhino-Laryngology 2017, 274,1231-1235. We like this classification because it puts accent on the basic features of choanal atresia – hypoplasia and atrophy.
Point 4: I do not think Table 2 is necessary to be shown in the manuscript since it is just reference data for the patients, not extracted data from the authors’ patients. Besides, the criteria are also a bit old. Is there any revision of the criteria?
Response 4: We have included in the publication all the data that enable to prove our results. Table 2 shows standard dimensions of the choanae and vomer according to the age groups of patients stated by Slovis et al. It was the aim of our study to compare the measurements we have from our patients with choanal atresia with the ones presented by Slovis et al. We think that it is very important to show this concrete table, it is our control group of healthy patients.
Point 5: The presentations of Figures 3-6 are disorganized and somehow confusing to readers. The presentation of the SD bar is different from the standard display of scientific articles. I strongly suggest that the authors reorganize Figures 3-6 and consider simplifying them into 2 figures.
Response 5: We have unified Table 3 and 4 into one figure. Figure 5 shows LWNC-V in patients with membranous choanal atresia. Figure 6 shows LWNC-V on the unaffected side in the observed group. These two figures cannot be joined together.
Point 6: Line 194-207, since this paper did not present any data regarding the gene anomalies regarding choanal atresia, I do not think the discussion of genopathy is necessary for this paragraph.
Response 6: We wanted to provide ENT doctors with a brief overview of the possible causes of choanal atresia. Kurosaka (2018) has done it very nice in his publication. It is important also from the clinical point of view – knowledge about some infuence of retinoic acid or thioamides administered during pregnancy is very useful.
Point 7: Line 267-273, as the main purpose of this paper, stated in the title is the value of applying nasal cavity CT; however, no data is presenting the surgical procedure related to CT findings. Therefore, I think it would be inappropriate to state surgical intervention in the discussion.
Response 7: In the discussion we have introduced the experience of IPOG members regarding current surgical interventions. We have modified our Conclusions and explained in a more detailed way the impact of our measured data on the surgical procedures we have been applying.
Submission Date
31 December 2020
Date of this review
12 Jan 2021 08:11:49
© 1996-2021 MDPI (Basel, Switzerland) unless otherwise stated